# Steering magnonic dynamics and permeability at exceptional points in a parity–time symmetric waveguide

Xi-guang Wang[1,2], Guang-hua Guo[1] & Jamal Berakdar [2✉]

Tuning the magneto optical response and magnetic dynamics are key elements in designing magnetic metamaterials and devices. This theoretical study uncovers a highly effective way of controlling the magnetic permeability via shaping the magnonic properties of coupled magnetic waveguides separated by a nonmagnetic spacer with strong spin–orbit interaction (SOI). We demonstrate how a spacer charge current leads to enhancement of magnetic damping in one waveguide and a decrease in the other, constituting a bias-controlled magnetic parity–time (PT) symmetric system at the verge of the exceptional point where magnetic gains/losses are balanced. We find phenomena inherent to PT-symmetric systems and SOI-driven interfacial structures, including field-controlled magnon power oscillations, non-reciprocal propagation, magnon trapping and enhancement as well as an increased sensitivity to perturbations and abrupt spin reversal. The results point to a new route for designing magnonic waveguides and microstructures with enhanced magnetic response.

[1] School of Physics and Electronics, Central South University, 410083 Changsha, China. [2] Institut für Physik, Martin-Luther Universität Halle-Wittenberg, 06120 Halle/Saale, Germany. ✉email: jamal.berakdar@physik.uni-halle.de

Nanomagnetism is the backbone of spin-based memories, data processing, and sensorics. In a generic magnet, the permeability, meaning the magnetic response to a weak external perturbation is governed by the behavior of the spin waves which are collective transverse oscillations (with their quantum termed magnon) around the ground state. Miniaturized magnonic logic circuits[1–6] and waveguides operated at low energy cost with negligible Ohmic losses were demonstrated as channels for magnon-based information transfer and processing. Geometric confinements, nanostructuring, and material design allow furthermore a precise spectral and dispersion shaping.

Here, using analytical methods and full micromagnetic simulations, we present predictions for a special type of magnonic waveguides that exhibit a gain-loss mechanism resulting in extraordinary magnonic and magneto-optical properties.

At zero bias, a normal-metal spacer sandwiched between two magnetic layers results in a Ruderman–Kittel–Kasuya–Yosida (RKKY) coupling between the magnetic layers which can be ferro or antiferromagnetic depending on the spacer layer thickness[7–9]. The RKKY interaction is the result of the spin sensitivity of the spacer's itinerant electron scattering from the two magnetic layers[10] causing a dependence of the formed standing electron wave on the magnetic states of the confining layers. Hence, for a nanoscale spacer, the RKKY coupling depends substantially on the spacer thickness and the relative orientations of the layers' magnetizations, as has been confirmed experimentally[11].

A further key feature exploited here is that for a DC-biased spacer with a strong spin–orbit coupling the charge current (cf. Fig. 1a) generates spin accumulations with opposite polarizations **T** at the boundaries to the magnetic layers[12–16]. The strength of the spin accumulations is quantified by the so-called spin-Hall angle. In turn, these spin accumulations exert torques on the magnetizations of the waveguides which are termed spin–orbit torques (SOTs) and have opposite signs at the two spacers/magnetic layer interfaces. An interesting feature is that when the layers' magnetizations are excited as to generate magnons, SOT adds to the intrinsic damping of magnetic dynamics, enhancing or decreasing it, as detailed below (cf. for instance Eq. (1)). It is this key feature that serves to achieve a case where magnetic losses in one waveguide are balanced by antidamping in the other waveguide, constituting so a typical setting of a PT-symmetric system, as realized for instance in optical waveguides[17–22].

Originally, PT-symmetry in the present context was addressed for a mechanical system governed by the Hamiltonian $\hat{H} = \hat{p}^2/2m + V_R(\hat{r}) + i\lambda V_I(\hat{r})$ (where $\hat{r}$ and $\hat{p}$ are position and momentum operators) with the real functions $V_R$ and $V_I$ describing the generally complex potential. PT symmetry requires that $V_R$ is symmetric (even) while $V_I$ must be antisymmetric (odd) under a parity operation for $[PT, \hat{H}] = 0$ to be valid[23–25]. For $\lambda = 0$, the Hamiltonian is Hermitian and the spectrum is real. For a finite $\lambda$, meaning a non-Hermitian but PT-symmetric $H$ the spectrum remains real as long as $\lambda$ is below a certain threshold $\lambda_{th}$[23–25]. At this critical point $\lambda_{th}$, referred to as the exceptional point (EP) and beyond it, the emergence of finite imaginary parts of the eigenvalues of $H$ marks the phase transition from the exact to a broken PT-symmetry phase. Most experimental demonstrations for this phase transition were so far for optical systems[18–22,26–31]. Considering photonic waveguide, for instance, the guided modes may well be described with a Schrödinger-type wave-equation with the role of $V_R$ (and $V_I$) being played by the refraction index gradients across the waveguide materials and the cladding. By optical materials, engineering power loss may be introduced to control $V_I$. In our case, it is the bias-induced charge current that drives the system to the EP very precisely and without material modifications. The nanoscale dispersion shaping

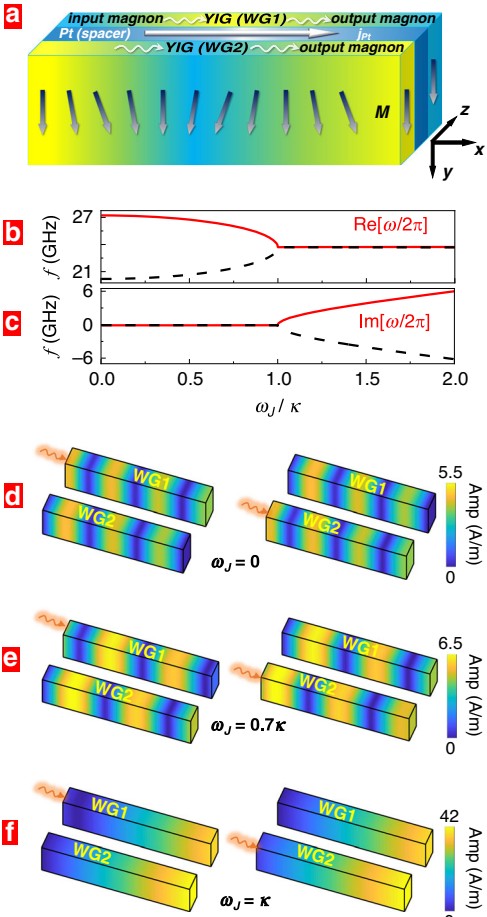

**Fig. 1 PT symmetry. a** Two magnetic layers (for example, YIG films) serve as magnonic waveguides labeled WG1 and WG2. The interlayer coupling is mediated by a metallic spacer with a large spin-Hall angle (for instance, Pt spacer). Driving a charge current $\mathbf{j}_{Pt}$ along the spacer ($x$ direction) results in spin-Hall torques acting on the magnetic waveguides. The torques damp or antidamp the magnetic dynamics in WG1 and WG2 resulting so in a PT-symmetric structure that exhibits special features of magnon propagation and magnetic permeability. Magnon wave packets launched locally at one end of WG1 or WG2 (left side in the figure) are steered, amplified, or suppressed by external fields that tune the waveguides from the PT-symmetric to the PT-symmetry broken phase at the exceptional point, where magnetic losses in WG1 balance magnetic antidamping in WG2. This is achieved for instance, by changing the ratio between the waveguides' coupling strength $\kappa$ and the magnitude of the spin-Hall torques $\omega_J$. **b** Real and **c** imaginary parts of two magnon eigenmode frequencies $f = \omega/(2\pi)$ as we scan $\omega_J/\kappa$ for the wave vector $k_x = 0.1$ nm$^{-1}$. **d–f** Spatial profiles of propagating spin-wave amplitude for a different loss/gain balance. The color change from blue to red corresponds to a linear amplitude change ranging from 0 to the maximum input signal. The local microwave field excites spin waves at the left side of the waveguide and has a frequency of 20 GHz. The length (along $x$ axis) of waveguides in **d–f** is 580 nm.

of magnons is an additional advantageous feature of PT-symmetric magnonics. The PT-symmetric magnonic waveguides proposed here are accurately tunable by external field parameters without any permanent material changes. As evidenced by explicit simulations and explained analytically within simplified models. The predicted phenomena are generic and appear for metallic as well as for insulating waveguides and do not require special precise tuning of the various internal interactions and material parameters. The dynamics can be readily

driven from the linear to the non-linear regime and the external fields such as the voltage bias can be pulsed, allowing so to study the magnon dynamics due to the temporal appearance or disappearance (quenching) of the EP. In addition to the documented advantages of magnons, the theoretical results point to functionalities that can be integrated in optomagnonic, spintronics, and magnonic circuits.

It should be noted, that PT-symmetry was also studied in other fields such as optomechanics[32,33], acoustics[34,35], and electronics[36–39]. PT symmetry in magnetism was theoretically discussed for macrospins involving balanced loss (due to intrinsic damping) and gain (imparted by parametric driving or spin-transfer torques)[40–43]. An experiment study was conducted on layers with graded intrinsic damping[44]. Further reports are on cavity magnon-polaritons involving phonon dissipation or electromagnetic radiation and parametric driving or SOT effects[45–47].

## Results and discussion

**Spin-torque-driven PT-symmetric waveguides.** Technically, our magnon signal propagates along the **x** direction in two planar parallel magnetic waveguides (the waveguide plane defines the $x - y$ plane). The two magnetic waveguides are coupled via the RKKY exchange interaction (cf. Fig. 1a). A charge current flowing in a spacer with a large spin-Hall angle (such as Pt) generates a SOT $\mathbf{T}_1\|\mathbf{y}$ on first waveguide enhancing the effective damping, and a SOT $\mathbf{T}_2\| -\mathbf{y}$ on second waveguide weakening the effective damping. The spin density accumulated at the spacer/magnetic layers boundaries derives from $\mathbf{T}_1 = \mathbf{z} \times \mathbf{j}_{Pt}$ in WG1, and $\mathbf{T}_2 = (-\mathbf{z}) \times \mathbf{j}_{Pt}$ in WG2 where $\mathbf{j}_{Pt}$ is the charge current density in the spacer. In a generic ferromagnet and for the long-wavelength spin excitations of interest here, to describe the magnetic dynamics it is sufficient to adopt a classical continuous approach and solve for the equations of motion of the magnetization vector fields $\mathbf{M}_p(\mathbf{r}, t)$ ($p = 1, 2$ enumerates the two waveguides), which amounts to propagating the Landau–Lifshitz–Gilbert (LLG) equation[12–15],

$$\frac{\partial \mathbf{M}_p}{\partial t} = -\gamma \mathbf{M}_p \times \mathbf{H}_{\text{eff},p} + \frac{\mathbf{M}_p}{M_s} \times \left[ \alpha \frac{\partial \mathbf{M}_p}{\partial t} + \gamma c_J \mathbf{T}_p \times \mathbf{M}_p \right]. \quad (1)$$

The waveguides are located at $z = +z_0$ and $z = -z_0$, and $\alpha$ is the conventional Gilbert damping inherent to magnetic losses in each of the waveguides. $\gamma$ is the gyromagnetic ratio. Let us define the unit vector field $\mathbf{m}_p = \mathbf{M}_p/M_s$, where $M_s$ is the saturation magnetization. The effective field $\mathbf{H}_{\text{eff},p} = \frac{2A_{\text{ex}}}{\mu_0 M_s} \nabla^2 \mathbf{m}_p + \frac{J_{\text{RKKY}}}{\mu_0 M_s t_p} \mathbf{m}_{p'} + H_0 \mathbf{y}$ consists of the internal exchange field, the interlayer RKKY coupling field, and the external magnetic field applied along the $y$ axis, where $p, p' = 1, 2$, and $p' \neq p$. $A_{\text{ex}}$ is the exchange constant, $J_{\text{RKKY}}$ is the interlayer RKKY exchange coupling strength, $t_p$ is the thickness (along $z$ axis) of the $p$th layer film (lying in the $x - y$ plane), and $\mu_0$ is the vacuum permeability. Of key importance to this study is the strength $c_J = T\theta_{\text{SH}} \frac{\hbar J_e}{2\mu_0 e\, t_p M_s}$ of SOT, which is proportional to the charge-current density $J_e$ and the spin-Hall angle $\theta_{\text{SH}}$ in the spacer layer. For instance, at an exceptional point discussed below we use, $c_J = 1 \times 10^5$ A/m corresponds to a charge current density of $J_e = 9 \times 10^8$ A/cm² in Pt[16]. $T$ is the transparency at the interface, and $e$ is the electron charge. Our proposal applies to a variety of settings, in particular synthetic antiferromagnets[48] offer a good range of tunability. To be specific, we present here numerical simulations for Pt interfaced with a Yttrium–Iron–Garnet (YIG) waveguides as experimentally realized for instance in ref. [16] corresponding to the following values $M_s = 1.4 \times 10^5$ A/m, $A_{\text{ex}} = 3 \times 10^{-12}$ J/m (technical details of the numerical realization and settings for other materials including metallic waveguides, are in the Supplementary Information). For the Gilbert damping, we

use $\alpha = 0.004$, but note that depending on the quality of the waveguides $\alpha$ can be varied within a wide range, down to two orders of magnitude smaller. The interlayer exchange constant is $J_{\text{RKKY}} = 9 \times 10^{-5}$ J/m², which is in the range for typical materials[49]. The interlayer coupling is more pronounced for a spacer thickness in the range of 5 nm. As established, $J_{\text{RKKY}}$ oscillates with the spacer thickness[10]. Here, we mainly focus on the case with ferromagnetic coupling $J_{\text{RKKY}} = 9 \times 10^{-5}$ J/m² corresponding to an exchange field of $\frac{J_{\text{RKKY}}}{\mu_0 M_s t_p} \approx 1 \times 10^5$ A/m. Experimentally, fabrication of a certain spacer thickness is well under control and allows so to tune to certain $J_{\text{RKKY}}$. The results for different $J_{\text{RKKY}}$ are discussed in Supplementary Information where also the role of further magnetic interaction such as dipolar fields is discussed. Below, we consider a multilayered waveguide film thickness of $t_{1,2} = 4$ nm (along $z$ axis) and assume that a sufficiently strong magnetic field $H_0 = 2 \times 10^5$ A/m is applied in plane along the $+y$ direction to drive the WGs to the remnant state. Equation (1) describes the linear and the non-linear transversal spin excitations; below we illustrate the non-linear dynamics by full numerical simulations. Before that, it is instructive to consider the linear (small transversal excitations) regime which allows for direct analytical insight into the PT-behavior of our system.

**Magnonic coupled wave-guide equations with spin–orbit torque.** For an interpretation of the full-fledge numerical simulations presented below let us formulate a simplified analytical model by considering small deviations of $\mathbf{m}_{s,p} = (\delta m_{x,p}, 0, \delta m_{z,p})$ away from the initial equilibrium $\mathbf{m}_{0,p} = \mathbf{y}$. Introducing $\psi_p = \delta m_{x,p} + i\delta m_{z,p}$, we deduce from linearizing Eq. (1) the coupled waveguide equations

$$\begin{aligned} i\frac{\partial \psi_1}{\partial t} - [(\omega_0 - \alpha\omega_J) - i(\omega_J + \alpha\omega_0)]\psi_1 + q\psi_2 &= 0, \\ i\frac{\partial \psi_2}{\partial t} - [(\omega_0 + \alpha\omega_J) + i(\omega_J - \alpha\omega_0)]\psi_2 + q\psi_1 &= 0. \end{aligned} \quad (2)$$

For convenience, we introduce in addition to the coupling strength $q = \frac{\gamma J_{\text{RKKY}}}{(1+i\alpha)\mu_0 M_s t_p}$, the SOT coupling at zero intrinsic damping $\kappa = \gamma J_{\text{RKKY}}/(\mu_0 M_s t_p) = q|_{\alpha\to 0}$. The intrinsic frequency of the waveguides is given by $\omega_0 = \frac{\gamma}{1+\alpha^2}(H_0 + \frac{2A_{\text{ex}}}{\mu_0 M_s}k_x^2 + \frac{J_{\text{RKKY}}}{\mu_0 M_s t_p})$, which is for the material studied here is in the GHz ($k_x$ is wavevector along the $x$ direction). The SOT-dependent gain-loss term $\omega_J = \frac{\gamma c_J}{1+\alpha^2}$ is essential for the PT-symmetry-related effects that we discussed here. Equation (2) admits a clear interpretation: the magnonic guided modes in the first waveguide (WG1) are subject to the confining complex potential $V(z) = V_R(z) + iV_I(z)$ with $V_R(z_0) = \omega_0 - \alpha\omega_J$ and $V_I(z_0) = -\omega_J - \alpha\omega_0$. In WG2, the potential is $V_R(-z_0) = \omega_0 + \alpha\omega_J$ and $V_I(-z_0) = \omega_J - \alpha\omega_0$. The mode coupling is mediated by $q$, which determines the periodic magnon power exchange between WG1 and WG2 in absence of SOT.

As outlined in the introduction, for a PT-symmetric system the condition $V_R(z_0) = V_R(-z_0)$ and $V_I(z_0) = -V_I(-z_0)$ applies, which is obviously fulfilled if the intrinsic damping is very small ($\alpha \to 0$). Comparing the current and the photonic case, in the latter the sign of the imaginary part of the WGs refractive indices (effectively acting as the light confining potentials) is tuned. Here we control with SOT the imaginary part of the permeability. This finding is important for the design of gyrotropic and (PT-symmetric) magneto-optical materials. For an explicit demonstration, we give below the expressions for the magnetic susceptibility (cf. also the Supplementary Information). Hence, the cases discussed here point to a promising route for designing PT-symmetric magneto-photonic structures via permeability engineering. We note, for a finite magnetic damping $\alpha$ a PT-behavior is still viable as confirmed by the full numerical

simulations that we discuss below and in the Supplementary Information.

## Magnon dynamics across the PT-symmetry breaking transition.
The dispersion $\omega(k_x)$ of the modes governed by Eq. (2) reads

$$\omega = (1 - i\alpha)\omega_0 \pm \sqrt{q^2 - \omega_J^2 + 2i\alpha\omega_J^2 + \alpha^2\omega_J^2}, \quad (3)$$

which describes both the acoustic and optical magnon modes[50] and depends parametrically on $\omega_J$ and $q$. For $\alpha \to 0$ (in which case $q \equiv \kappa$), the eigenvalues are always real in the PT-symmetric regime below the gain/loss-balance threshold $\omega_J/\kappa < 1$. At the exceptional point $\omega_J/\kappa = 1$, the two eigenvalues and eigenmodes become identical. For $\omega_J/\kappa > 1$ (by increasing the current density for instance), we enter the PT-symmetry broken phase, and the eigenvalues turn complex, as typical for PT-symmetric systems[31,51]. The splitting between the two imaginary parts is determined by $2\kappa[(\omega_J/\kappa)^2 - 1]^{1/2}$ and is tunable by external fields. This fact is useful when exploiting the enhanced waveguides sensitivity, meaning achieving an enlarged magnetic response to relatively small magnetic perturbations. Allowing for a small damping $\alpha$ does not alter the modes behavior, as demonstrated by the full numerical results in Fig. 1b, c. The full magnon dispersions (Re[$\omega$] versus $k_x$ curves) for $\omega_J/\kappa < 1$ and $\omega_J/\kappa = 1$ are shown in Fig. 2. The symmetry of our waveguides brings in a special behavior of the magnon signal transmission, meaning the propagation of a superposition of eigenmodes: Without charge current in the spacer ($\omega_J = 0$), a signal injected at one end in one waveguide oscillates between WG1 and WG2 (due to the coupling $\kappa$) in a manner that is well-established in coupled waveguide theory (cf. Fig. 1d). Exciting two magnon modes (symmetric acoustic mode and antisymmetric optical mode) with different finite $k_x$, the interference between the two modes leads to periodic exchange of energy between the two waveguides. The coupling length is dependent on the wave vector difference of the two modes[2,20,51]. The interference pattern results in a spatially non-uniform spin-wave amplitude. Switching on the charge current, $\omega_J/\kappa$ becomes finite and the beating of the magnon power between WG1 and WG2 increases (cf. Fig. 1e), as deducible from Eq. (2), and also encountered in optical waveguides[51]. Equation (2) also indicates that near the exceptional point, a magnonic wavepacket injected in one waveguide no longer oscillates between the two waveguides but travels simultaneously in both waveguides, as confirmed in Fig. 1f by full numerical simulations. Passing EP ($\omega_J/\kappa > 1$) the magnonic signal always propagates in the waveguide with gain and is quickly damped in the waveguide with loss. Some of our observations resemble those in optical systems[20]. There, the confining potential for an electromagnetic wave in a PT-symmetry study is realized typically by material engineering of the spatial distribution of the complex refractive-index such that the real part is spatially symmetric and the

imaginary part is antisymmetric with respect to an exchange of the waveguides. Experimental sophisticated deposition techniques of multilayered structure[20,31] can imprint the desired variation of the dielectric function. A magnonic realization offers a work-around without a permanent material modification; by just changing the voltage at the ends of Pt wire and tuning the current density to the specific ratio $\omega_J/\kappa$ the EP vicinity is approached. This flexibility allows not only for easy control of the PT-symmetry-related features but also offers a way to study the dynamics of EP-quench by applying voltage pulses that switch off and on the EP. An example is shown below. Furthermore, the influence of an external magnetic field adds yet another knob to change the ratio $\omega_J/\kappa$. Generally, we find a non-reciprocal propagation below the exceptional point.

## Enhanced sensing at PT-symmetry breaking transition and PT-dependent permeability.
To study the behavior of permeability across the PT-phase transition, we derive expressions for the magnetic susceptibility, i.e., the linear response of our setup to external magnetic perturbations. To this end, we apply an external microwave field $\mathbf{h}_p$, which adds to the effective field in the LLG equation. In frequency space, we deduce that $\widetilde{\psi}_p = \sum_{p'} \chi_{pp'} \, \gamma \widetilde{h}_{m,p'}$ (tilde stands for Fourier transform, more details are in Supplementary Information), with $h_{m,p} = h_{x,p} + ih_{z,p}$, and $\chi_{pp'}$ is the dynamic magnetic susceptibility which has the matrix form

$$\chi = \frac{1}{(\omega_k - i\alpha\omega - \omega)^2 + \omega_c^2 - \kappa^2} \begin{pmatrix} (\omega_k - i\alpha\omega) + (i\omega_c - \omega) & \kappa \\ \kappa & (\omega_k - i\alpha\omega) - (i\omega_c - \omega) \end{pmatrix}, \quad (4)$$

with $\omega_c = \gamma c_J$ and $\omega_k = \gamma(H_0 + \frac{2A_{ex}k_x^2}{\mu_0 M_s} + \frac{J_{RKKY}}{\mu_0 M_s t_p})$.

Near the exceptional point, the system becomes strongly sensitive, for instance to changes in the charge current term $\omega_c$, as testified by the behavior of the susceptibility. Figure 3 demonstrate this behavior for the imaginary parts of $\chi_{11}$ and $\chi_{12}$. The high sensitivity of the excited spin waves on $\omega_c$ near the exception point (Fig. 3c–f) is exploitable to detect slight changes in charge current density $c_J$. Furthermore, near the exceptional point, our setup is strongly sensitive to changes in the magnetic environment. For example, at $\omega_J = \kappa$ if the magnetic field $H_0$ (or local magnetization) is increased by 100 A/m in WG1, large-amplitude spin-wave oscillations are generated in WG2, as evidenced by the time dependence of $M_x(x = 2000\text{ nm})$ in WG2 (Fig. 3g). The spin-wave amplification leads eventually to a reversal of $M_y$ in WG2 (Fig. 3h). The increased sensitivity is PT-symmetry-related, in fact, away from the PT-breaking transition, e.g., for $\omega_J = 0.7\kappa$, when $H_0$ is reduced by the same amount in WG1, virtually no changes in propagating spin waves are observed (not shown). Obviously, this magnon amplification may serve as a tunable sensor for the magnetic environment.

## Current-induced switching in magnetic PT-symmetric junctions.
A special feature of magnetic systems (in contrast to optical waveguides, for instance) is the possibility of current-induced magnetization switching (described by Eq. (1) but not by Eq. (2))[52].

In fact, for large current densities, we are well above the exceptional point. In this case, the magnetic system becomes unstable against switching. We find, with further increasing the charge current density (enhancing $\omega_J$) the local magnetization in waveguide 2 is indeed switched to $-y$. Magnon dynamics above the exceptional point is still possible however by tuning the spacer

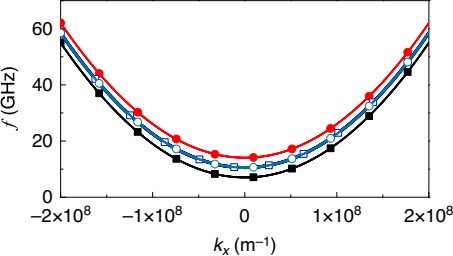

**Fig. 2 Magnon dispersion.** Merging of the acoustic ($\omega_J = 0$, solid squares) and optical magnon ($\omega_J = 0$, solid circles) modes dispersion Re[$\omega$]($k_x$) when approaching the loss/gain-balance (exceptional) point $\omega_J = \kappa$ (open dots).

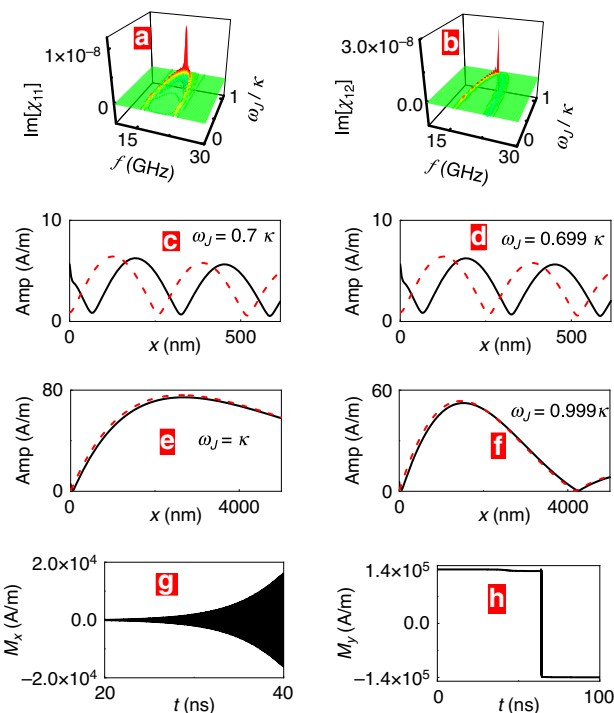

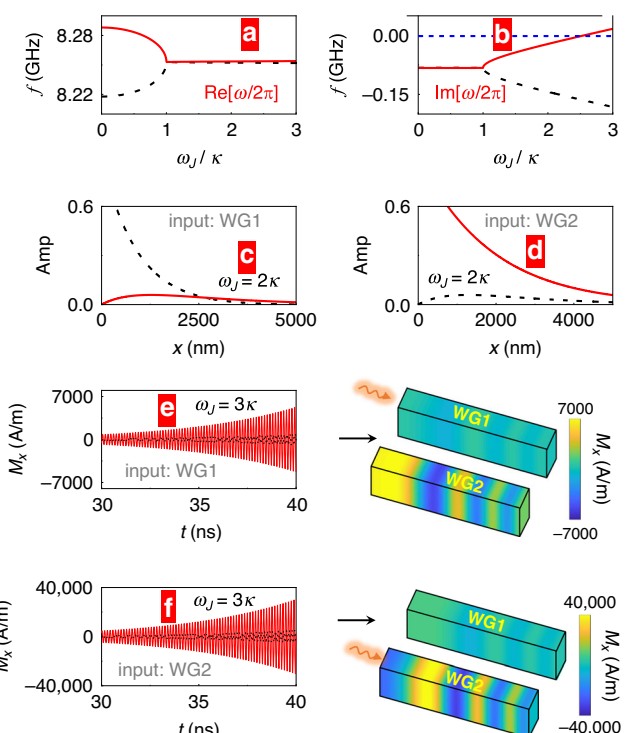

**Fig. 3 Enhanced sensing.** Magnetic susceptibility Im[$\chi_{11}$] (**a**) and Im[$\chi_{12}$] (**b**) as functions of $f$ and $\omega_J$. Peaks in Im[$\chi_{11}$] and Im[$\chi_{12}$] are found at the exceptional point $\omega_J = \kappa$. **c–f** Exciting spin waves of frequency 20 GHz at $x = 0$ in WG1, spatial profiles of spin-wave amplitude for different $\omega_J$. Black solid line and red dashed line represents the amplitudes in WG1 and WG2, respectively. Near $\omega_J = \kappa$, a slight variation in $\omega_J$ causes marked changes in the spin-wave amplitudes (**e**, **f**), while the change is negligible near $\omega_J = 0.7\kappa$ (**c**, **d**). **g**, **h** At $\omega_J = \kappa$, increasing the external magnetic field ($H_0 = 2 \times 10^5$ A/m) in WG1 by 100 A/m, the time dependence of $M_x$(**g**) and $M_y$(**h**) at $x = 2000$ nm in WG2.

**Fig. 4 Magnon amplification. a**, **b** Real and imaginary parts of the eigenmodes $\omega$ as varying the loss/gain balance by scanning $\omega_J$ (meaning, the SOT strength). The wave vector is $k_x = 0.03$ nm$^{-1}$ and the intrinsic coupling between WG1 and WG2 $\kappa$ is lowered, as compared to Fig. 1 (by choosing $J_{RKKY} = 9 \times 10^{-7}$ J/m$^2$). **c**, **d** Spatial profiles of magnon wave amplitudes (as normalized to their maxima) for $\omega_J = 2\kappa$, and (Re[$\omega$] = $2\pi \times 20$ GHz). Black dashed line corresponds to WG1 and red solid line to WG2. **e**, **f** Time dependence (at the location $x = 2000$ nm) and the spatial profiles (at $t = 40$ ns) of the $x$ component of the magnetization $M_x$ for $\omega_J = 3\kappa$. The color variation from blue to red corresponds to a $M_x$ change from the negative maximum to the positive maximum.

material properties or its thickness to obtain a smaller $\kappa$, for instance with $J_{RKKY} = 9 \times 10^{-7}$ J/m$^2$ and $\alpha = 0.01$. In this case the condition $\omega_J \gg \alpha\omega_0$ is not satisfied anymore, and the influence of intrinsic magnetic losses ($\alpha$) in both waveguides are important. Nonetheless, even without reaching the strict PT-symmetric condition, we still observe that the real parts of the two eigenvalues merge at the same point $\omega_J = \kappa$, and the two imaginary parts become different when $\omega_J > \kappa$, as shown by Fig. 4a, b. We elaborate on the influence of varying $\alpha$ in Supplementary Information. When $\omega_J = 2\kappa$, the two imaginary parts are both negative, meaning that both modes are evanescent. The propagation of the magnonic signal launched in one waveguide's end is shown in Fig. 4c–d evidencing that the spin waves in both waveguides decay differently. Here, the non-uniform profile of spin-wave oscillation indicates a finite wave-vector $k_x$. An input signal in the waveguide with enhanced damping leads to an evanescent spin-wave in WG1. Injecting the signal in WG2, the attenuation of the spin-wave is weaker, and its amplitude is always larger. When $\omega_J = 3\kappa$ and Im[$\omega$] of the optical magnon mode turns positive, we observe that SOT induces a spin-wave amplification with time (Fig. 4e, f). This finding is interesting for cavity optomagnonics[53].

For input signal in WG1 or WG2, the spin-wave amplitude is always larger in WG2 with the negative effective damping. Also, the excited spin-wave amplitude is much larger when the input is in the WG2. Thus, no matter from which waveguide we start, the output signal is always distributed at the end of WG2, a fact that can be employed for constructing magnonic logic gates.

**Dynamic control of PT symmetry and exceptional point quenching.** Since the EP and the PT-symmetric behavior are triggered by an external voltage and considering that the typical time scale for the magnetic dynamics is in the nano to picoseconds[54], it is well experimentally feasible to pulse the external voltage on this time scale which leads to an on/off switching of the EP-influenced dynamics. Figure 5 demonstrates such a case where we apply a sequence of charge current pulses with a period of 50 ns. Switching on the charge current with $\omega_J = \kappa$, magnons at EP propagate simultaneously in two waveguides, and the magnon amplitude is larger due to the enhancement in magnetic susceptibility. When the charge current is switched off, the PT symmetry is restored, and the magnon amplitude drops (Fig. 5a, b). Aside from the charge current, changing the equilibrium magnetization by applying an external magnetic field also allows controlling the PT-symmetry-affected dynamics. For example, applying $H_0 = 2 \times 10^5$ A/m along $+x$, SOT does not influence the effective damping, and thus effects due to PT symmetry are off and magnons with a smaller amplitude oscillate between the two waveguides (Fig. 5c). Furthermore, as demonstrated in ref. [54], the magnetic anisotropy can also be affected by voltage pulses leading to yet another way to modify the magnonic dynamics around the EP. Besides, switching and tuning the charge current to have the system slightly above the EP (for example, $\omega_J = 1.05\kappa$), the local magnetization in WG2 can be switched to $-y$. Thus, pulsing the external voltage can periodically switch between the parallel and anti-parallel state in both waveguides

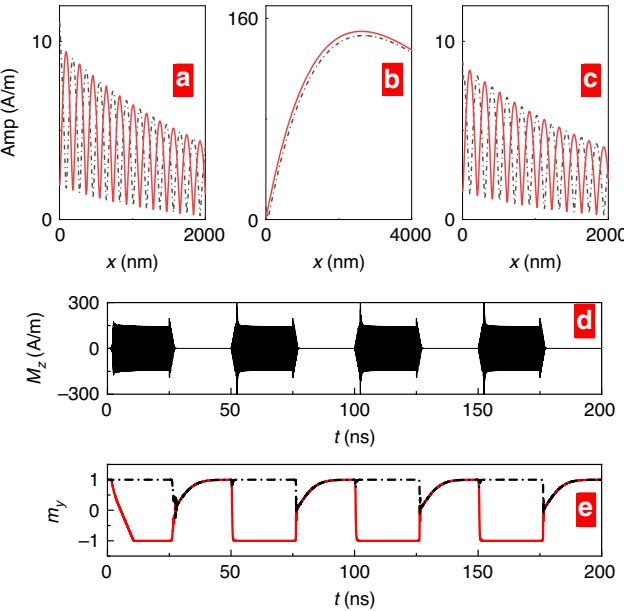

**Fig. 5 EP quenching.** When exciting spin waves of frequency 20 GHz at $x = 0$ in WG1, the spatial profiles of spin-wave amplitudes are shown for **a** $\omega_J = 0$, **b** $\omega_J = \kappa$ with $H_0 = 2 \times 10^5$ A/m applied along $+y$, and **c** $\omega_J = \kappa$ with $H_0 = 2 \times 10^5$ A/m applied along $+x$. Black dashed line and red solid line represent the amplitudes in WG1 and WG2, respectively. **d** Time dependent of $M_z$ is detected at $x = 3000$ nm in WG1 when the charge current $\omega_J$ is periodically switched on ($\omega_J = \kappa$) and off ($\omega_J = 0$). **e** Time dependent of $m_y$ of WG1 and WG2 when the charge current $\omega_J$ is periodically switched on ($\omega_J = 1.05\kappa$) and off ($\omega_J = 0$).

(Fig. 5e), which is of relevance to magnetic tunnel junctions whose magneto conductance depends on the relative orientation of the layers' magnetizations.

**Dzyaloshinskii–Moriya interaction in electrically controlled PT-symmetric waveguides.** In magnetic layers and at their interfaces a special type of exchange called Dzyaloshinskii–Moriya (DM) interaction[55,56] may exist. The DM interaction is an anti-symmetric interaction induced by spin–orbit coupling due to the broken inversion symmetry in lattices or at the interface of magnetic films. In the context of our work, it is particularly interesting that the DM interaction may allow for coupling to an external electric field **E** and to voltage gates. The contribution to the system free energy density in the presence of DM and **E** is $E_{elec} = -\mathbf{E} \cdot \mathbf{P}$, with the spin-driven polarization $\mathbf{P} = c_E[(\mathbf{m} \cdot \nabla)\mathbf{m} - \mathbf{m}(\nabla \cdot \mathbf{m})]$[57,58]. This alters the magnon dynamics through the additional term $\mathbf{H}_{elec} = -\frac{1}{\mu_0 M_s}\frac{\delta E_{elec}}{\delta \mathbf{m}}$ in the effective field $\mathbf{H}_{eff}$. To uncover the role of DM interaction on the magnon dynamics in PT-symmetric waveguides, we consider three cases:

(i)  The two waveguides experience the same static electric field $\mathbf{E}_{1,2} = (0, 0, E_z)$.
(ii)  The electric fields in both waveguides are opposite to each other, i.e., $\mathbf{E}_1 = (0, 0, E_z)$ and $\mathbf{E}_2 = (0, 0, -E_z)$.
(iii)  The electric field is applied only to waveguide 1. These situations are realized by appropriate local gates.

For the case (i) with $\mathbf{E}_{1,2} = (0, 0, E_z)$, $\omega_0 = \frac{\gamma}{1+\alpha^2}(H_0 - \frac{2c_E E_z k_x}{\mu_0 M_s} + \frac{2A_{ex}}{\mu_0 M_s}k_x^2 + \frac{J_{RKKY}}{\mu_0 M_s t_p})$ in Eq. (2), and the condition for PT-symmetry still holds. Applying an electric field along the $z$ axis causes an asymmetry in the magnon dispersion. As shown by Fig. 6, a positive $E_z$ shifts the dispersion towards positive $k_x$ while a negative $E_z$ shifts it in the opposite direction. With increasing $\omega_J$,

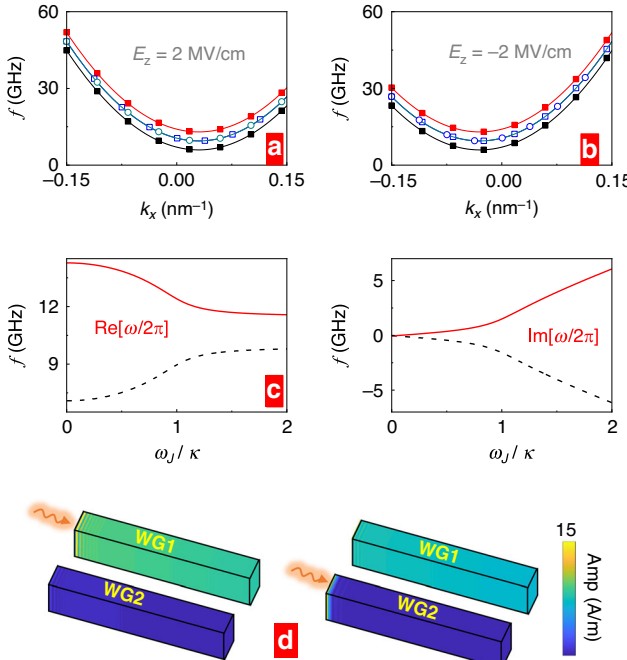

**Fig. 6 Influences of DM interaction.** Control of the coupled magnonic waveguides characteristics by external electric fields in the presence of Dzyaloshinskii–Moriya interaction. **a**, **b** Applying a static electric field $\mathbf{E}_{1,2} = (0, 0, E_z)$ with $E_z = \pm 2$ MV/cm to both waveguides modifies the magnon dispersion $Re[\omega](k_x)$ curves for $\omega_J = 0$ (solid dots) and $\omega_J = \kappa$ (open dots). **c** Real and imaginary parts of the two eigenmodes $\omega$ as functions of $\omega_J$ and in the presence of two static electric fields (or voltages) applied with opposite polarity to both waveguides ($\mathbf{E}_1 = (0, 0, E_z)$ and $\mathbf{E}_2 = (0, 0, -E_z)$, and $E_z = 2$ MV/cm) at $k_x = 0.1$ nm$^{-1}$. **d** Spatial profiles of the propagating spin-wave amplitudes when applying the electric field to WG1 ($\mathbf{E}_1 = (0, 0, E_z)$ with $E_z = 2$ MV/cm, and $\mathbf{E}_2 = (0, 0, 0)$). Color scale from blue to red corresponds to an amplitude change from 0 to its maximum.

the changes of $Re[\omega]$ and $Im[\omega]$ (not shown) are similar to what has been observed in Fig. 1b, c.

As for the case $\mathbf{E}_1 = (0, 0, E_z)$ and $\mathbf{E}_2 = (0, 0, -E_z)$, in the two equations (2) $\omega_0$ is different. Explicitly: $\omega_0 = \frac{\gamma}{1+\alpha^2}(H_0 \mp \frac{2c_E E_z k_x}{\mu_0 M_s} + \frac{2A_{ex}}{\mu_0 M_s}k_x^2 + \frac{J_{RKKY}}{\mu_0 M_s t_p})$ where the $-$ sign applies for WG1 and the $+$ sign corresponds to WG2. Hence, under an asymmetric electric field, the potential $V_R$ is not even ($V_R(z_0) \neq V_R(-z_0)$), and the PT-symmetry condition can not be satisfied. The $\omega_J$ dependence of $Re[\omega]$ and $Im[\omega]$ are shown in Fig. 6, and no exceptional point can be strictly identified in this case.

For case (iii), we set $\mathbf{E}_1 = (0, 0, E_z)$ and $\mathbf{E}_2 = (0, 0, 0)$. The PT-symmetry condition is not satisfied. When the electric field is applied only to a single waveguide, it shifts selectively the magnon dispersion relation in this waveguide. Therefore, the magnon wave in the lower frequency range propagates solely in the waveguide with the electric field. As shown in Fig. 6d, we excite the magnonic wavepacket with a frequency in the WG1 or WG2, the magnonic wave always propagates in the waveguide 1 which amounts to a magnon channeling by an electric field, while the propagation in the other waveguide is suppressed. This example illustrates yet another handle to steer magnonic waves swiftly and at low energy consumption by pulsed electric gating.

**Conclusions.** Magnonic waveguides based on magnetic junctions can be designed to exhibit a transition from a PT-symmetric to a

PT-symmetry broken phase. Extraordinary magnetic response and magnon propagation features are predicted near the transition (exceptional) point that makes such devices interesting for use in magnonic logics, as effective sensors for changes in the magnetic environments, as magnonic amplifiers, or for applications based on magnetic switching such as magnetic tunnel junctions. The setup exhibits enhanced magnetic susceptibility near the PT-phase transition, which is reflected in the permeability (cf. Supplementary Information) pointing to a route to PT-symmetry-assisted magneto-photonics. Magnonic propagation is highly controllable by external electric and magnetic fields that derive the system across the exceptional point which render possible a controlled power distribution in the waveguides, as well as a non-reciprocal propagation or amplified magnon waves. An existing DM interaction allows for dispersion engineering and modifies the PT-symmetry-related feature in a way tunable by external electric fields. No permanent material change is needed to achieve the PT-symmetry phase transition. Changing external field parameters drives the system to the exceptional point and allows for precise control of the spectral position and the strength of the PT-symmetry breaking. Furthermore, using pulsed fields offers a route for studying EP-quench dynamics and a time-resolved PT-symmetry breaking transition. The presented results underline the potential of PT-symmetry-induced magnonics as the basis for additional functionalities of magneto-photonic, spintronics, and cavity magnonic devices that are highly controllable by external parameters.

## Data availability

All technical details for producing the figures are enclosed in the supplementary information. Data are available from the authors upon request.

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

## Acknowledgements

This research is financially supported by the DFG through SFB 762 and SFB TRR227, the National Natural Science Foundation of China (Nos. 11704415, 12074437, 11674400, 11374373), and the Natural Science Foundation of Hunan Province of China (No. 2018JJ3629).

## Author contributions

X.-g.W. performed all numerical simulations and analytical modeling. J.B. conceived and supervised the project. X.-g.W. and J.B. wrote the paper. X.-g.W., J.B., and G.-h.G. discussed and agreed on the content.

## Funding

## Competing interests

The authors declare no competing interests.
