## [Peer Review File · Nature Communications]

Reviewers' Comments:

Reviewer #1:

Remarks to the Author:

This is a theoretically oriented work. Authors propose a new approach to manipulate the magnon propagation in magnonic devices based on PT symmetry-related processes. This is a relatively new topic, and the ideas in this work would enrich the PT symmetry applications in the field of Magnetism/Magnonics.

There is nothing wrong with this work, but I am not sure that it belongs to a high profile journal such as Nature Communications, since it does not present a substantive advance in the field. In addition, I have a few comments and questions for the authors' consideration.

1. The magnonic gain in this work is achieved by spin orbit torque (SOT) from the Pt interlayer. In this case and PT symmetry may be reached when the damping in the two FM waveguides is very weak, as in YIG. Can the authors comment about the influence of the damping value in the FM waveguide if it is not negligible?
2. In Eq. (1) with SOT, the authors only consider the damping term of the SOT. How about regular damping in the FM; will it affect the conclusion in this work?
3. RKKY interaction between two FM substances varies from positive to negative values, and vice versa depending on the distance between the two FMs. Have the authors considered the situation of negative RKKY interaction in their analysis?
4. Non-uniform spin waves in ferromagnet are formed by magnet size confinement. It is not clear what is the generation mechanism of non-uniform spin waves in the model presented here.
5. Authors consider two YIG waveguides with small damping and RKKY interaction. In reality, RKKY interaction usually exists in ferromagnetic metals such as permalloy, and they are much easier for device fabrications compared with that of YIG. Can authors add some discussion about the case of metallic FM? This would broaden the application of this work for more abundant FMs.

Reviewer #2:

Remarks to the Author:

In general, I like the ideas reported in the manuscript and I believe that the potential devices based on the new approach to steering magnetisation dynamics would be very useful. However, personally I am not satisfied with the level of explanations provided in the manuscript, and also I found several unclear discussions that in my opinion would prevent readers from understanding the article. The authors should extend the discussion by focusing more closely of the physics of the effects exploited in the work. I encourage them to consider the following suggestions to improve their manuscript.

1. In my opinion, the Introduction section should provide at least a basic explanation of the physical effects such as 'Ruderman-Kittel-Kasuya-Yosida (RKKY) interaction', 'spin orbit torques (SOTs)', and 'Dzyaloshinskii-Moriya interaction'. Although an expert should understand 'A hallmark of PT-symmetric systems is that, even if the underlying Hamiltonian is non-Hermitian, the eigenvalues may be real (...)', it is highly desirable to write at least one paragraph with a formula to facilitate the discussion. Last but not least, the concept of PT-symmetry is not introduced at all. I agree that this concept is well-known in optics and acoustics, and the authors cite the key papers, but a good article should be self-consistent and it should think of non-experts such as students.
2. The authors solve the LLG equations and consider several limiting cases using material parameters of suitable and well-known materials such as YIG and Pt. Whereas this approach is beneficial, I find it to be too 'mathematical' for a paper that reports novel physical effects and devices. For example, the authors write 'The interlayer exchange constant $J_{RKKY} = 9 \times 10^{-5}$ J/m², which is in the typical range.' Yes, I would agree. However, what is the physical meaning of this constant? Moreover, the waveguide thickness is just 4 nm. I wonder if it feasible (I also note

that the waveguide dimensions are not indicated in Fig. 1a).

3. The choice of the Gilbert damping constant and its discussion are unclear. The authors choose $\alpha = 0.004$, also noting that in practice it can be much smaller. Then, on p. 7 they write 'Allowing for a small damping α does not alter the modes behavior'. Yet, from the Supplementary Information it follows that α is not very important for their analysis. The authors should explain this better.

4. The discussion around the text 'This behavior resembles the optics case. We note that in our waveguides, this limit is simply achieved by tuning the external electric and magnetic fields that then change the ratio $\omega J / \kappa$.' The authors erroneously assume that their readers would be experts in optics. Whereas talking through the 'optics case' should help, a good discussion of how the aforementioned control of the electric and magnetic fields can be realised has to be provided in the manuscript, because I would say that this would be technologically challenging.

5. The authors also find a regime of non-reciprocal propagation below the exceptional point and write about this in the manuscript. Thus, they should explain this in more detail in the manuscript.

6. The results in Fig. 3 are for the frequency of 20 GHz and above. Is it possible to operate at a lower frequency? From our experience with FMR spectroscopy measurements, the operation at frequencies above 20 GHz is challenging (high losses, noise, etc.)

7. At some point the authors claim that their results 'point to a new route to designing optomagnonic waveguides, cavities and structures with enhanced magneto-optical response, and sensors with enhanced sensitivities'. This is not discussed in the manuscript at all. This discussion must be provided.

8. Unfortunately, I found quite a few typos and technically awkward phrases. For example, 'low-energy magnetic response', 'separated by current carrying spacer', 'control on magnon', 'emergent magnetic properties', 'perturbations round the exceptional point', and '(authors) discussed, interpret and agreed on the content'.

9. In the Supplementary Information, the authors write that they excite spin waves by applying a local microwave magnetic field. The problem of efficient spin wave excitation is important and therefore I encourage the authors to provide more technical details.

Reviewer #3:

Remarks to the Author:

The paper by Wang, Guo and Berakdar considers spin wave dynamics in a magnetic waveguide structure that they call a "parity-time (PT) symmetric waveguide". The structure represents two magnetic electrically insulating "layers" separated by a non-magnetic metallic spacer with strong spin-orbit interaction (SOT). Unfortunately, comparing Fig. 1(a) and the system description in the middle of Page 5, I got slightly confused. The figure seems to show a planar structure similar to the "side-coupled magnetic stripes" from Ref. [6] in the manuscript. However, while describing the system (Middle of Page 5), the authors speak of "layers" with particular values of "thicknesses". Furthermore, the form of the inline expression for $H_{eff,p}$ on that page, for instance the $1/t_p$ dependence of the "interlayer" exchange field suggests that in reality, the authors deal with a YIG/Pt/YIG tri-layer film with infinite in-plane directions and plane spin waves propagating in the film plane.

Therefore I decided to be guided by the wording from page 5, and, in the following, I assume that the authors deal with a tri-layer film as described above.

By itself, the idea that a microwave PT waveguide can be implemented employing the tri-layer structure and SOT is novel and very interesting. The authors suggest a theoretical model to describe the spin wave dynamics of the system and carry out numerical simulations to confirm the validity of their concept of a magnon-based microwave PT waveguide. Although the discussion of the proposed geometry is very short, as most often in theory papers, I do not see obstacles that cannot be overcome while implementing the proposed concept experimentally, however I believe that experimentalists will reconsider it significantly while adopting the author's idea to available film-fabrication and magnonic-device fabrication and characterisation techniques. From this point of view, this concept may attract interest of the broad Magnonics community, and once the concept has been confirmed with an experiment, the present theoretical paper may become of interest to a very broad audience.

What I am not happy with, is the oversimplification of the model. The authors ignore completely the contribution of the dipole-dipole interaction on the dynamics, although the thickness of the magnetic layers is relatively large - 4 nm, which implies that the total thickness of the magnetic material is 8 nm. This is a very usual approach that is valid while constructing abstract models for magnon dynamics. However, I do not believe that it is appropriate for a paper with a focus on real-world applications. I understand why the authors ignored the dipole field - if it were included, the model would become much more complicated, potentially essentially 2-dimensional, and it would not be that easy to reduce Eq.(1) to Eqs.(2) and (3) in the manuscript. However, the dipole field contribution to the properties of spin waves in the films with thicknesses in the range 8-10 nm can be significant, see e.g. Fig. 1 in Stashkevich et al., Phys. Rev. B 91, 214409 (2015).

Given that, in order for the paper to be considered as appropriate for publication in Nat. Comm., the authors should either include the dynamic dipole field in their theory or provide a strong argument why this contribution can be neglected for the considered numerical example. Otherwise, this paper is fully appropriate for publication in a journal with more focus on abstract physical concepts and aimed at experts in the narrower field, such as Physical Review B.

I also suggest that the authors reconsider Fig. 1 and its description in view of my critique above.

Response to the comments of Reviewer #1

Dear Reviewer,

thank you very much for assessing our work. Please, find below a point-to-point response. Changes and additions to the revised version and the supplemental information are marked blue.

Comment: *This is a theoretically oriented work. Authors propose a new approach to manipulate the magnon propagation in magnonic devices based on PT symmetry-related processes. This is a relatively new topic, and the ideas in this work would enrich the PT symmetry applications in the field of Magnetism/Magnonics.*

There is nothing wrong with this work, but I am not sure that it belongs to a high profile journal such as Nature Communications, since it does not present a substantive advance in the field. In addition, I have a few comments and questions for the authors' consideration.

Answer: We appreciate the referee's remark regarding the novelty and usefulness of our results for the magnetism/magnonic research. In fact, the work benefited substantially from general discussions with colleagues in the photonic community regarding the challenges in controlling precisely and dynamically the losses in photonic waveguides without physical material insertion or permanent change in the dielectric response. In the present scheme, the spectral position of the exceptional point (EP) and the strength of the PT symmetry breaking are controlled by simply tuning external fields such as the bias voltage that drives the charge current in the spacer layer. This is not a mere technical advance, but also allows to address new fundamental aspects: For an illustration we added additional calculations to the new version showing how current pulses (pulsed bias voltage) switch the EP off and on temporally in a precisely controlled manner, allowing so to study the dynamics around the EP [page 17 and new Fig. 5 in manuscript]. On the spintronic side, the technical relevance to magnetic switching and enhanced magnetic response are now discussed in more details. A further feature inherent to our system is the presence of a wide range of spin-related interactions (such as the Dzyaloshinskii–Moriya interaction (DMI)) that allow for instance to engineer the intrinsic dispersions spatio-temporally (by acting with local external fields, for example) leading to new phenomena such as those discussed in the new version of the paper and the supplementary information (SI).

A further aspect regarding magnetic waveguides is the presence of dipolar interactions

and the inherent non-linear nature of the equations governing the spin dynamics, new simulations are added to highlight both issues. Indeed, we believe that the results have the potential to trigger new type of magneto-phonic studies that involve ingredients from magnonics/spintronics and photonics. As demonstrated by the calculations of the permeability, the magnetic response can be substantially enhanced in a controlled way, which should also affect the propagation of electromagnetic waves in these waveguides. From a technical point of view, magnons have several favorable features such as the nano-scale wave-length, low energy dissipation, and ready integration in spintronic circuits. We believe that these and further well-established properties of magnons are very beneficial for a possible experimental realization and for boosting theory and experimental studies on PT-symmetry.

Comment: 1. *The magnonic gain in this work is achieved by spin orbit torque (SOT) from the Pt interlayer. In this case and PT symmetry may be reached when the damping in the two FM waveguides is very weak, as in YIG. Can the authors comment about the influence of the damping value in the FM waveguide if it is not negligible?*

Answer: To address this issue specifically and to make calculation-supported conclusions we performed detailed additional calculations that are now added as a new section to the SI [pages 2-5 and Fig. S1 in SI]. We kindly refer to this part of the work to avoid repetition here. As evidenced by these calculations and their analysis the main conclusions of the paper are valid for a substantially large range of magnetic damping.

Comment: 2. *In Eq. (1) with SOT, the authors only consider the damping term of the SOT. How about regular damping in the FM; will it affect the conclusion in this work?*

Answer: In SI we added the results for the case of a stronger magnetic damping [pages 3-5 in SI]. As inferred from the results of the full-fledge numerical calculations, the conclusions of the linearized model with a smaller magnetic damping are robust and are still valid. We note however that both the linear regime as well as YIG films with a very low damping have already been realized experimentally, so the linearized model is also relevant for experiments.

Comment: 3. *RKKY interaction between two FM substances varies from positive to negative values, and vice versa depending on the distance between the two FMs. Have the*

authors considered the situation of negative RKKY interaction in their analysis?

Answer: Thank you for bringing up this point. We performed detailed calculations with varying type and strength of the RKKY interaction and expanded the SI accordingly [Pages 7-8 and Figs. S3-S4 in SI]. We also added a detailed discussion of the RKKY interaction to the main text. Our conclusion is that as long as the RKKY interaction is effective in coupling the waveguides, the PT-symmetry behavior persists. We note that magnetic multilayer structures of our interest can be fabricated with impressive accuracy.

Comment: 4. *Non-uniform spin waves in ferromagnet are formed by magnet size confinement. It is not clear what is the generation mechanism of non-uniform spin waves in the model presented here.*

Answer: The spin waves studied here have a finite wave-vector k_x . By exciting two spin wave modes (symmetric acoustic mode and antisymmetric optical mode in the two waveguides) that have the same frequency and different k_x , the interference between the two modes leads to a periodic transfer of the energy between the two waveguides, generating so a spatially non-uniform spin wave profile, as shown in Fig. 1(d). A similar behavior has been observed experimentally in magnonic systems [Q. Wang, P. Pirro, R. Verba, A. Slavin, B. Hillebrands and A. V. Chumak, *Sci. Adv.* 4, e1701517 (2018)], in quantum systems [S. Klaiman, U. Günther and N. Moiseyev, *Phys. Rev. Lett.* 101, 080402 (2008)], and optical systems [C. E. Rüter, K. G. Makris, R. El-Ganainy, D. N. Christodoulides, M. Segev and D. Kip, *Nat. Phys.* 6, 192 (2010).]. Increasing ω_f causes k_x of the two modes to move towards each other, and results in a change of the relative phase in the two waveguides making the two modes asymmetric. This brings about a non-uniform non-reciprocal wave propagation (Fig. 1(e)). At the EP, the two spin wave modes coalesce at the same k_x , travelling simultaneously in the two waveguides (Fig. 1(f)).

We added pertinent remarks to the manuscript. Further discussions and results are included in SI [Page 4 in SI].

Comment: 5. *Authors consider two YIG waveguides with small damping and RKKY interaction. In reality, RKKY interaction usually exists in ferromagnetic metals such as permalloy, and they are much easier for device fabrications compared with that of YIG. Can authors add some discussion about the case of metallic FM? This would broaden the application of this work for more abundant FMs.*

Answer: Thank you for this advice. We constructed and analyzed the case of coupled permalloy waveguides and added these new materials to the SI. For metallic coupled waveguides, the electric current also flows in the FM layer, and the corresponding in-plane spin transfer torque has to be accounted for in this case, which we have done. As evidenced by the results of numerical simulations included now in the SI, the main conclusions still hold also for such a metallic system [Pages 8-10 and Fig. S5 in SI], which in fact widens the scope of this study.

Response to the comments of Reviewer #2

Dear Reviewer,

thank you very much for your time in studying our work. Below you find a point-to-point response to the report. Changes and additions to the revised version and the supplemental information are marked blue.

Comment: *In general, I like the ideas reported in the manuscript and I believe that the potential devices based on the new approach to steering magnetisation dynamics would be very useful. However, personally I am not satisfied with the level of explanations provided in the manuscript, and also I found several unclear discussions that in my opinion would prevent readers from understanding the article. The authors should extend the discussion by focusing more closely of the physics of the effects exploited in the work. I encourage them to consider the following suggestions to improve their manuscript.*

Answer: We thank the referee for his/her valuable advice for improving the presentation and discussion of the results. We studied all comments and suggestions and amended the manuscript accordingly.

Comment: *1. In my opinion, the Introduction section should provide at least a basic explanation of the physical effects such as 'Ruderman-Kittel-Kasuya-Yosida (RKKY) interaction', 'spin orbit torques (SOTs)', and 'Dzyaloshinskii-Moriya interaction'. Although an expert should understand 'A hallmark of PT-symmetric systems is that, even if the underlying Hamiltonian is non-Hermitian, the eigenvalues may be real (...)', it is highly desirable to write at least one paragraph with a formula to facilitate the discussion. Last but not least, the concept of PT-symmetry is not introduced at all. I agree that this concept is well-known in optics and acoustics, and the authors cite the key papers, but a good article should be self-consistent and it should think of non-experts such us students.*

Answer: Thank you for drawing our attention to this weak point of the presentation. The introductory section has been completely rewritten such that the discussed concepts and phenomena from magnetism, optics, and mathematics are introduced for a generally interested reader. We also explained on this level the goals of the study and the relevance

of the achieved results.

Comment: 2. The authors solve the LLG equations and consider several limiting cases using material parameters of suitable and well-known materials such as YIG and Pt. Whereas this approach is beneficial, I find it to be too 'mathematical' for a paper that reports novel physical effects and devices. For example, the authors write 'The interlayer exchange constant $J_{\text{RKKY}} = 9 \times 10^{-5} \text{ J/m}^2$, which is in the typical range.' Yes, I would agree. However, what is the physical meaning of this constant? Moreover, the waveguide thickness is just 4 nm. I wonder if it is feasible (I also note that the waveguide dimensions are not indicated in Fig. 1a).

Answer: The RKKY coupling J_{RKKY} is introduced in the new version in more details. We mentioned briefly the physics behind it as well as the meaning of the involved parameters and added several references. In particular, we indicated the oscillatory behavior of J_{RKKY} as a function of the spacer thickness and how its strength decays for thicker spacer. Several references from theory and experiment are added. In the context of this current study, the effects related to the PT symmetry are well-pronounced for a spacer thickness in the range of 5 nm. From an experimental point of view, the spacer thickness can indeed be adjusted to produce a suitable behavior of J_{RKKY} . In the main text, we focus mainly on the case of a ferromagnetic coupling $J_{\text{RKKY}} = 9 \times 10^{-5} \text{ J/m}^2$, and a corresponding exchange field in the range of $J_{\text{RKKY}} / (\mu_0 M_s t_p) \approx 1 \times 10^5 \text{ A/m}$. The new version contains detailed discussions and additional calculations for different J_{RKKY} (added in to the SI). Also, further references to experiments on RKKY coupling between multilayers are added. Experimentally, thin YIG film can be prepared to be as thin as 4 nm. [J. C. Gallagher, A. S. Yang, J. T. Brangham, B. D. Esser, S. P. White, M. R. Page, K.-Y. Meng, S. Yu, R. Adur, W. Ruane, S. R. Dunsiger, D. W. McComb, F. Yang and P. C. Hammel, Appl. Phys. Lett. 109, 072401 (2016); B. Heinrich, C. Burrowes, E. Montoya, B. Kardasz, E. Girt, Y.-Y. Song, Y. Sun and M. Wu, Phys. Rev. Lett. 107, 066604 (2011)]. The thickness (along z axis) of our magnetic layers is 4 nm. The length (along x axis) of the model is 40 μm which rules out spurious effects due to spin wave reflections from geometric boundaries. The film extends along the y axis. Related discussions and necessary modifications are added to the new version.

Comment: 3. The choice of the Gilbert damping constant and its discussion are unclear. The authors choose $\alpha = 0.004$, also noting that in practice it can be much smaller. Then, on p. 7 they write 'Allowing for a small damping α does not alter the modes

behavior'. Yet, from the Supplementary Information it follows that alpha is not very important for their analysis. The authors should explain this better.

Answer: Thank you for drawing our attention to this point. The supplementary information (SI) have been now modified to address the issue of damping in details with statements supported by full numerical results [Pages 2-5 in SI]. As evident from these calculations, the main conclusions of the paper are robust to damping in a wide range.

Comment: 4. *The discussion around the text 'This behavior resembles the optics case. We note that in our waveguides, this limit is simply achieved by tuning the external electric and magnetic fields that then change the ratio ω_J/κ . ' The authors erroneously assume that their readers would be experts in optics. Whereas talking through the 'optics case' should help, a good discussion of how the aforementioned control of the electric and magnetic fields can be realized has to be provided in the manuscript, because I would say that this would be technologically challenging.*

Answer: We modified the pertinent part of the paper to make the discussion more accessible and to outline the parallels/differences between the magnonic and photonic cases. Generally, in optics the potential for studying the PT-symmetry breaking transition is realized by a designed spatial distribution of the complex refractive-index (for instance the real part is spatially symmetric, but the imaginary part is antisymmetric with respect to a parity operation that exchanges the two waveguides). The experimental realization relies on an elaborate deposition technique for accomplishing the desired multilayered structure. In our waveguide, one simply changes the voltage at the end of the Pt spacer to ramp the electric current density (similar to the experiments on current-induced switching but in our case at lower current densities). By doing so one changes the key ratio ω_J/κ that takes the system in a very controlled manner to the EP. In addition, one may pulse the voltage and study the EP dynamics [Pages 17 and new Fig. 5 in manuscript]. We added to the new version additional simulation results and discussion to support these statements.

Comment: 5. *The authors also find a regime of non-reciprocal propagation below the exceptional point and write about this in the manuscript. Thus, they should explain this in more detail in the manuscript.*

Answer: The non-reciprocal wave propagation originates from the superposition of two

asymmetric magnon modes (caused by SOT). Without SOT, the superposition of the symmetric and the anti-symmetric magnon modes results in a reciprocal propagation. In SI, we added full details and discussions concerning the behavior of magnon modes and their properties [Page 4 in SI].

Comment: 6. *The results in Fig. 3 are for the frequency of 20 GHz and above. Is it possible to operate at a lower frequency? From our experience with FMR spectroscopy measurements, the operation at frequencies above 20 GHz is challenging (high losses, noise, etc.)*

Answer: We tested for this issue: With smaller RKKY coupling strength and an applied magnetic field, the PT-symmetric waveguide allows for magnons with a much lower frequency. For example, by using $J_{\text{RKKY}} = 9 \times 10^{-6} \text{ J/m}^2$ and $H_0 = 2 \times 10^4 \text{ A/m}$, we find that the device can be operated at 2 GHz. These results and discussions are added to SI [Page 7 and Fig. S3 in SI].

Comment: 7. *At some point the authors claim that their results 'point to a new route to designing optomagnonic waveguides, cavities and structures with enhanced magneto-optical response, and sensors with enhanced sensitivities'. This is not discussed in the manuscript at all. This discussion must be provided.*

Answer: We added more details on these issues. In brief, driving the system to the EP leads to a substantially enhanced magnetic susceptibility. This claim is supported by deriving explicit analytical expressions for the susceptibility and by numerical simulations for the non-linear case (results are incorporated in the text and the SI). This enhancement in the susceptibility is reflected in a corresponding behavior of the permeability, meaning that the propagation of electromagnetic waves near the EP will be affected by the magnetic response. Such magneto-photonics devices are less studied due to the generally very small magneto-optical response; to circumvent this problem cavity-optomagnonics is considered a possible solution. Enhancing the magnetic response by driving the system to the EP is obviously a new general alternative. Furthermore, since the enhancement in the susceptibility is related to a certain current density threshold, one may exploit this enhanced magnetic response to sense such a charge current threshold. We added more explanation to the text and the SI. In addition, we also added calculations demonstrating the enhanced sensitivities around the EP to slight changes in magnetic field, [page 5 in SI] which is

beneficial to sensorics.

Comment: 8. *Unfortunately, I found quite a few typos and technically awkward phrases. For example, 'low-energy magnetic response', 'separated by current carrying spacer', 'control on magnon', 'emergent magnetic properties', 'perturbations round the exceptional point', and '(authors) discussed, interpret and agreed on the content'.*

Answer: We have corrected the typos. Thank you.

Comment: 9. *In the Supplementary Information, the authors write that they excite spin waves by applying a local microwave magnetic field. The problem of efficient spin wave excitation is important and therefore I encourage the authors to provide more technical details.*

Answer: In the SI, we added more details on the excitation process and some introductory remarks on the experimental technique for the excitation of spin waves [Page 6 and Fig. S2 in SI]. Thank you very much indeed.

Response to the comments of Reviewer #3

Dear Reviewer,

thank you very much for the detailed report. Please, find below a point-to-point response. Changes and additions to the revised version and the supplemental information are marked blue.

Comment: *The paper by Wang, Guo and Berakdar considers spin wave dynamics in a magnetic waveguide structure that they call a "parity-time (PT) symmetric waveguide". The structure represents two magnetic electrically insulating "layers" separated by a non-magnetic metallic spacer with strong spin-orbit interaction (SOT). Unfortunately, comparing Fig. 1(a) and the system description in the middle of Page 5, I got slightly confused. The figure seems to show a planar structure similar to the "side-coupled magnetic stripes" from Ref. [6] in the manuscript. However, while describing the system (Middle of Page 5), the authors speak of "layers" with particular values of "thicknesses". Furthermore, the form of the inline expression for $H_{\text{eff},p}$ on that page, for instance the $1/t_p$ dependence of the "interlayer" exchange field suggests that in reality, the authors deal with a YIG/Pt/YIG tri-layer film with infinite in-plane directions and plane spin waves propagating in the film plane.*

Therefore I decided to be guided by the wording from page 5, and, in the following, I assume that the authors deal with a tri-layer film as described above.

Answer: Thank you for drawing our attention to this issue. Indeed, we are dealing with a tri-layer film. We produced as suggested, a new Fig. 1(a) for the multilayer structure and enhanced the description of the considered multilayer film in the paper.

Comment: *By itself, the idea that a microwave PT waveguide can be implemented employing the tri-layer structure and SOT is novel and very interesting. The authors suggest a theoretical model to describe the spin wave dynamics of the system and carry out numerical simulations to confirm the validity of their concept of a magnon-based microwave PT waveguide. Although the discussion of the proposed geometry is very short, as most often in theory papers, I do not see obstacles that cannot be overcome while implementing the proposed concept experimentally, however I believe that experimentalists*

will reconsider it significantly while adopting the author's idea to available film-fabrication and magnonic-device fabrication and characterisation techniques. From this point of view, this concept may attract interest of the broad Magnonics community, and once the concept has been confirmed with an experiment, the present theoretical paper may become of interest to a very broad audience.

Answer: Thank you for the encouraging remarks, especially concerning the experimental feasibility. Indeed, similar remarks we received from experimental colleagues in magnonics/spintronics, and also photonics. So, we are really hopeful that the study will generate quite some research activities in this direction.

Comment: *What I am not happy with, is the oversimplification of the model. The authors ignore completely the contribution of the dipole-dipole interaction on the dynamics, although the thickness of the magnetic layers is relatively large - 4 nm, which implies that the total thickness of the magnetic material is 8 nm. This is a very usual approach that is valid while constructing abstract models for magnon dynamics. However, I do not believe that it is appropriate for a paper with a focus on real-world applications. I understand why the authors ignored the dipole field - if it were included, the model would become much more complicated, potentially essentially 2-dimensional, and it would not be that easy to reduce Eq.(1) to Eqs.(2) and (3) in the manuscript. However, the dipole field contribution to the properties of spin waves in the films with thicknesses in the range 8-10 nm can be significant, see e.g. Fig. 1 in Stashkevich et al., Phys. Rev. B 91, 214409 (2015).*

Answer: We agree that the issue of the dipole-dipole interaction has to be clarified. To this end we implemented the dynamic dipolar field in the theory and carried out full numerical simulations, as well as analytical calculations and analysis. The results are summarized and presented in a new section in the supplementary information (SI) [pages 10-14 and Figs. S6-S7 in SI]. As evidenced by the new results, one can indeed conclude that the micromagnetic simulations with the dipolar fields follow the general predictions of the proposed analytical models for the magnonic PT-symmetric waveguides. In particular, the new results in the SI endorse the feasibility of a PT-symmetry breaking transition at EP. The simulations show that the dipolar field only slightly increases the ratio ω_I / κ at EP in the lower frequency range. In the high frequency range, the contribution of the dipolar field is weak, and the results are not much affected by this contribution.

Comment: *Given that, in order for the paper to be considered as appropriate for publication in Nat. Comm., the authors should either include the dynamic dipole field in their theory or provide a strong argument why this contribution can be neglected for the considered numerical example. Otherwise, this paper is fully appropriate for publication in a journal with more focus on abstract physical concepts and aimed at experts in the narrower field, such as Physical Review B.*

I also suggest that the authors reconsider Fig. 1 and its description in view of my critique above.

Answer: Thank you very much for the detailed and helpful advice to inspect the role of the dipolar interactions and on how to enhance the presentation. We hope that the PT-symmetry topic in general will trigger further ideas and concepts in the flourishing fields of magnonics and spintronics and also brings the fields of magnonic and photonics a step closer.

Reviewers' Comments:

Reviewer #1:

Remarks to the Author:

The authors have addressed my detailed comments point by point. However, I am still not convinced that this work merits publication in a high profile journal such as Nature Communications as I commented previously. The reasons are as following:

1. This work is not the first paper about PT symmetry in Magnetism/Magnonics. Two theory groups have already reported the possibility of achieving PT symmetry in Magnonics, such as PRB 91, 094416 (2015) (Ref. 40); PRB 94, 020408 (R) (2016) (Ref. 41); Phys. Rev. B 97, 201411(R) (2018) (Ref. 42); and Scientific Reports 9, 17484 (2019) [not mentioned by the present authors]. In addition, one experimental group has also reported PT in Magnonics (Science Advances 5 (11), eaax9144 (2019) [not reported by the present authors]. Also spin orbit torque induced PT symmetry was analyzed by Galda et al. (Ref. 42). Therefore, I consider the present work as an incremental advance.

2. New magnetic interactions have been added in this paper including RKKY and DM interactions. However, to my best knowledge there are no experiments that show evidence for these interactions in YIG/Pt system as dealt with in the present work.

3. The new added interactions certainly bring up new phenomena; but it is not clear which interaction is dominant. In addition SOT induced magnetization switching and non-reciprocal spin wave propagation via DM interaction are very well known processes in Magnetism; thus these are not unique in the present theoretical paper.

In conclusion, I recommend this paper be submitted to a more specific journal such as PRB.

Reviewer #2:

Remarks to the Author:

The authors satisfactorily addressed all comments and suggestions that I raised previously. The revised manuscript reads better than its original version. In particular, the new discussion of the impact of the dipole-dipole interaction strengthens the manuscript and makes the theoretical predictions more suitable for potential experimental confirmation works.

I have 3 minor comments:

1) In my opinion, the Introduction is written such that readers could perceive this paper as an experimental one. The authors should change this.

2) The entire Introduction is formatted as one large paragraph. This is stylistically incorrect.

3) In Supplementary Information, "Finite difference approximation is used" is redundant and also grammatically incorrect.

I think that after one more revision this paper can be accepted for publication in Nature Communications.

Reviewer #3:

Remarks to the Author:

The authors have addressed all comments by all reviewers in a constructive manner, and the paper can now be accepted for publication in Nature Communications.

The response to the comments of Reviewer #1

Dear Reviewer,

Thank you very much for your continuing interest in this work. Please, find below a point-to-point response. Changes in the new resubmission are marked blue.

Comment: *The authors have addressed my detailed comments point by point. However, I am still not convinced that this work merits publication in a high profile journal such as Nature Communications as I commented previously.*

Answer: We are glad that our previous point-to-point response regarding the content of the paper was satisfactory. We hope that the current response will clarify the differences to previous literature and the general importance of the new findings.

Comment: *The reasons are as following. 1. This work is not the first paper about PT symmetry in Magnetism/Magnonics. Two theory groups have already reported the possibility of achieving PT symmetry in Magnonics, such as PRB 91, 094416 (2015) (Ref. 40); PRB 94, 020408 (R) (2016) (Ref. 41); Phys. Rev. B 97, 201411(R) (2018) (Ref. 42); and Scientific Reports 9, 17484 (2019) [not mentioned by the present authors]. In addition, one experimental group has also reported PT in Magnonics (Science Advances 5 (11), eaax9144 (2019) [not reported by the present authors]. Also spin orbit torque induced PT symmetry was analyzed by Galda et al. (Ref. 42). Therefore, I consider the present work as an incremental advance.*

Answer: Indeed, our paper is not (and does not claim to be) the first paper discussing a possible occurrence of PT symmetry in magnetism. The above mentioned earlier papers are focused on how PT symmetry may occur in magnetism. Below, we address each paper separately in some details.

To the best of our knowledge, our work is the first to propose PT-symmetric magnonic waveguides with the new PT-symmetry-associated predictions, detailed in our paper. Beyond the novelty of the proposal, we see the merit of our work in the fundamental and technological importance of the revealed new phenomena that are shown to occur in an experimentally feasible structure as well as in the generic nature of the predictions. These statements follow from analysis of a general model system which we confirm by full numerical simulations for a variety of materials in an experimentally realistic setting. Furthermore, on the photonic side, we are not aware of any previous studies on non-

invasive PT-symmetric magnetophotonic structure, meaning a system whose permeability is steerable to the PT-symmetry phase transition by an applied voltage. Thus, we are convinced that the obtained results are indeed novel, important and of interest to the magnetic and photonic communities as well as for researchers with interest in PT symmetry.

To summarize some of the new findings: The paper reports on extraordinary magnetic response and magnon propagation features, including the control of magnon power oscillations, non-reciprocal spin-wave propagation profile near the exception point (EP), magnon trapping and enhancement at EP with enhanced sensing, and time-dependent switching of the EP. Also, our study uncovers the role of further interactions that influence the magnonic dynamics (DMI and dipolar interactions) and exposes the related phenomena with explicit numerical simulations and analytical expressions. The inclusion of these interactions is important for the predictive power of the theory providing thus a strong ground for an experimental realization.

Differentiation to the papers mentioned in the report:

- Paper [PRB 91, 094416 (2015) (Ref. 40)] points out that PT symmetry may occur in macroscopic magnetic structures due to balanced gain and loss. The authors suggest realizing the loss by intrinsic damping in one macrospin, and gain by parametric driving or spin-transfer torque (STT) in the other macrospin.

In contrast to PRB 91, 094416 (2015), in our paper we discuss magnonic waveguides from a general point of view and concretely for experimentally feasible structures and introduce new mechanisms for steering the system to the PT-symmetry breaking point. Furthermore, we uncover genuinely new propagation, excitations, and control phenomena are uncovered and interpreted using simplified analytical models and confirmed for experimentally realistic systems with full-fledge numerical simulations.

- Papers [PRB 94, 020408 (R) (2016) (Ref. 41); Phys. Rev. B 97, 201411(R) (2018) (Ref. 42)] discuss a balanced loss (from intrinsic damping) and gain (from STT) in macrospin model and a chain model. The authors point out the appearance of PT symmetry in these systems. The proposal is to encircle the transition point using a specially-designed time-dependent magnetic field for realizing a conversion between different eigenmodes [cf. also Scientific Reports 9, 17484 (2019)].

Again, the PT-symmetric magnonic waveguides presented in our paper and the variety

of phenomena in a realistic setting were NOT addressed NOR the subject of these previous papers [PRB 94, 020408 (R) (2016) (Ref. 41); Phys. Rev. B 97, 201411(R) (2018) (Ref. 42), Scientific Reports 9, 17484 (2019), appeared Nov. 25. 2019].

- Paper [Science Advances 5 (11), eaax9144 (2019), appeared Nov. 22. 2019] is an experimental work on PT-symmetry via coupling two magnets with different intrinsic damping (meaning a loss-loss situation), the existence of the PT symmetry is verified. This is fundamentally different from our waveguide magnonic case where the gain-loss is externally tunable allowing for the various predictions presented in the paper.

In the revised version, we referenced and commented upon the missing papers, including those works that appeared during or after our submission (started Nov. 12. 2019).

We wish to emphasize that our work points to genuinely new and experimentally feasible effects. In particular, the predicted phenomena are robust to intrinsic damping in a very wide range. The PT symmetry is swiftly switchable on and off via an electric voltage, by electric pulses or by reversing the local magnetization. Besides, we provide evidence that the new effects related to PT symmetry and EP remain unchanged within a large wave-vector and frequency range. Furthermore, by inspecting the permeability we suggest to use the structure as a PT-symmetric magnetophotonic metamaterial. In addition, in the context of cavity optomagnonics, we demonstrate the magnon local enhancement in magnitude and coupling strength to external fields by utilizing the PT-symmetry behavior of the proposed waveguides. Finally, we demonstrated the generic nature of the findings by presenting results for insulating and metallic structures as well as for systems where additionally other interactions may be operational (such as DM or/and, dipolar interactions as well as spin transfer torques, in additions to the spin-orbit torque).

Comment: 2. *New magnetic interactions have been added in this paper including RKKY and DM interactions. However, to my best knowledge there are no experiments that show evidence for these interactions in YIG/Pt system as dealt with in the present work.*

Answer: We may refer to the experimental observations [PRL 107, 066604 (2011); PRL 120, 127201 (2018)] about interlayer exchange coupling in YIG layer / ferromagnetic layer. The couplings used in our numerical simulations are in line with these experiments. As mentioned above the phenomena, reported in the paper are generic. In fact, we also

observed the PT-symmetry related effects in the structure depicted in figure R1. However, in view of the extensive evidence presented in the paper and in the supplemental materials we opted not to add yet another example.

Figure R1. Two magnetic waveguides (labeled as WG1 and WG2) are coupled via interlayer exchange coupling. Further attaching heavy metal layers on two waveguides, the induced spin-Hall torques damp or antidamp the magnetic dynamics in WG1 and WG2 resulting in PT symmetric structure.

In this context we recall, that in our previous response to Reviewer #1's report, we constructed and analyzed the case of permalloy waveguides in supplementary information (Pages 8-10 and Fig. S5), and our main conclusions still hold for such a metallic system (RKKY interaction may also exist in such system).

DMI in YIG originates from the magneto-electric coupling attributed to YIG [PRL 106, 247203 (2011); PRL 113, 037202 (2014) (Ref. 56)]. The magneto-electric coupling allows to control the magnetic dynamics through the electric field by generating an effective DMI, where the effective DM constant is $D = c_E |E|$. Alternatively, one can exploit the interfacial type DMI at the interfaces of magnetic layers to other typical material systems [PRL 104, 137203(2010) (Ref. 53); PRL 114, 047201(2015); Nat. Commun. 6, 7635 (2015)]. Thus, we see no principle or material-related obstacles in realizing the proposed setting.

Comment: 3. *The new added interactions certainly bring up new phenomena; but it is not clear which interaction is dominant. In addition SOT induced magnetization switching and non-reciprocal spin wave propagation via DM interaction are very well known processes in Magnetism; thus these are not unique in the present theoretical paper.*

Answer: The merit of the paper is certainly not in the introduction of SOT-induced magnetization switching nor in exposing the DMI-affected spin wave dispersion. We agree, these are indeed established effects. What is new here is the transition from a PT-symmetry conserving to a PT-symmetry-broken phase at the EP (Fig. 5) which brings in qualitatively different DMI-induced phenomena (Fig. 6), as discussed in the paper. SOT is used as a tool to control the PT symmetry behavior of the waveguides (not for conventional magnetic switching).

Regarding the roles of the relative strengths of interactions:

a) Which interaction is dominant depends on the materials of the waveguides. For example, DMI usually exists in some special structures [PRL 104, 137203(2010) (Ref. 53); PRL 114, 047201(2015); Nat. Commun. 6, 7635 (2015)], and RKKY coupling depends on the spacer thickness [Phys. Rev. B 44, 7131(1991)]. In our manuscript, the influences of different types of DMIs on the PT symmetry are discussed along with the role of the relative strength of RKKY coupling.

b) As demonstrated by several examples and Figures in the body of the paper and in the supplemental materials, the predicted phenomena do not require a special tuning of the relative strengths of interactions but exist in a wide range of parameters of the external fields and for different material classes.

We added a note on these facts to the revised paper.

We hope with the above arguments, we were able to explain why we strongly believe that Nature Communications is the right venue for this work and thank you for your kind and helpful comments.

The response to the comments of Reviewer #2

Dear Reviewer,

thank you very much for your efforts in assessing our work. We are glad to see that the revisions addressed satisfactorily the raised comments. We did further amendments and improved the text according to the second report. Changes in the new resubmission are marked blue.

Comment: *The authors satisfactorily addressed all comments and suggestions that I raised previously. The revised manuscript reads better than its original version. In particular, the new discussion of the impact of the dipole-dipole interaction strengthens the manuscript and makes the theoretical predictions more suitable for potential experimental confirmation works. I have 3 minor comments:*

Comment: *1. In my opinion, the Introduction is written such that readers could perceive this paper as an experimental one. The authors should change this.*

Answer: We emphasized the theoretical nature of the article in the introduction.

Comment: *2. The entire Introduction is formatted as one large paragraph. This is stylistically incorrect.*

Answer: Thank you for this hint. The Introduction has been separated in several paragraphs.

Comment: *3. In Supplementary Information, "Finite difference approximation is used" is redundant and also grammatically incorrect.*

Answer: The redundant sentence is deleted.

Comment: I think that after one more revision this paper can be accepted for publication in Nature Communications.

Answer: We thank you again for your very helpful advice and comments.

The response to the comments of Reviewer #3

Dear Reviewer,

Thank you very much indeed for your comments and advice in enhancing the paper.

***Comment:** The authors have addressed all comments by all reviewers in a constructive manner, and the paper can now be accepted for publication in Nature Communications.*

Reviewers' Comments:

Reviewer #1:

Remarks to the Author:

The authors have satisfactorily answered my comments. The manuscript is ready now for publication in Nature Communications.

Reviewer #2:

Remarks to the Author:

My understanding of this situation is that Referee #1 does not think that the present manuscript is suitable for publication in the Journal based on its high impact and esteem by the research community. However, Referee #3 recommends the paper for publication. Whereas the final decision will be made by the Editor, I would like to clarify my position because it seems to be important for the final decision. Initially, I did not really like the manuscript but I saw the potential of the ideas presented in it. The authors did a good job by revising the manuscript and addressing all comments in great detail. Thus, in my opinion, the current version of the manuscript can be recommended for publication. However, I am mindful of the opinion of Referee #1 who considers this work as an incremental advance. Actually, I would agree with this opinion. However, I do not think that this is an obstacle. Indeed, according to the webpage of the Journal, "Papers published by the journal represent important advances of significance to specialists within each field." I think that this present manuscript and the results presented in it satisfy this condition, i.e. 'important advance'. Hence, I would like to recommend it for publication. However, the authors must properly reference all papers mentioned in Referee's #1 report. This will allow the research community to establish the real importance of the findings reported in the manuscript. -- Sincerely,...

The response to the comments of Reviewer #1

Comment: *The authors have satisfactorily answered my comments. The manuscript is ready now for publication in Nature Communications.*

Dear Reviewer,

thank you very much indeed for all your comments and advice for enhancing the paper.